# Differential Carbon Catabolite Repression and Hemicellulolytic Ability among Pathotypes of *Colletotrichum lindemuthianum* against Natural Plant Substrates

**DOI:** 10.3390/jof10060406

**Published:** 2024-06-05

**Authors:** Karla Morelia Díaz-Tapia, María Guadalupe Zavala-Páramo, Maria Guadalupe Villa-Rivera, Ma. Irene Morelos-Martínez, Everardo López-Romero, June Simpson, Jeni Bolaños-Rebolledo, Horacio Cano-Camacho

**Affiliations:** 1Centro Multidisciplinario de Estudios en Biotecnología, FMVZ, Universidad Michoacana de San Nicolás de Hidalgo, Km 9.5 Carretera Morelia-Zinapécuaro, Posta Veterinaria, Morelia 58000, Michoacán, Mexico; morelia.diaz@umich.mx (K.M.D.-T.); irene.morelos@umich.mx (M.I.M.-M.); jeni.jbo@gmail.com (J.B.-R.); 2Escuela Nacional de Estudios Superiores, Unidad Morelia, Universidad Autónoma de México, Antigua Carretera a Pátzcuaro No. 8701, Morelia 58190, Michoacán, Mexico; gvillarivera@gmail.com; 3Departamento de Biología, División de Ciencias Naturales y Exactas, Universidad de Guanajuato, Noria Alta SN, Guanajuato 36030, Guanajuato, Mexico; everlope@ugto.mx; 4Centro de Investigación y Estudios Avanzados del Instituto Politécnico Nacional, Unidad Irapuato, Km 9.6 Libramiento Norte Carretera Irapuato-León, Irapuato 36821, Guanajuato, Mexico; june.simpson@cinvestav.mx

**Keywords:** *Colletotrichum lindemuthianum*, pathotypes, hemicellulases, cellulases, natural substrates, secretomes

## Abstract

*Colletotrichum lindemuthianum* is a phytopathogenic fungus that causes anthracnose in common beans (*Phaseolus vulgaris*) and presents a great diversity of pathotypes with different levels of virulence against bean varieties worldwide. The purpose of this study was to establish whether pathotypic diversity is associated with differences in the mycelial growth and secretion of plant-cell-wall-degrading enzymes (PCWDEs). We evaluated growth, hemicellulase and cellulase activity, and PCWDE secretion in four pathotypes of *C. lindemuthianum* in cultures with glucose, bean hypocotyls and green beans of *P. vulgaris*, and water hyacinth (*Eichhornia crassipes*). The results showed differences in the mycelial growth, hemicellulolytic activity, and PCWDE secretion among the pathotypes. Glucose was not the preferred carbon source for the best mycelial growth in all pathotypes, each of which showed a unique PCWDE secretion profile, indicating different levels of carbon catabolite regulation (CCR). The pathotypes showed a high differential hemicellulolytic capacity to degrade host and water hyacinth tissues, suggesting CCR by pentoses and that there are differences in the absorption and metabolism of different monosaccharides and/or disaccharides. We propose that different levels of CCR could optimize growth in different host tissues and could allow for consortium behavior in interactions with bean crops.

## 1. Introduction

The ascomycete *Colletotrichum lindemuthianum* is a fungus that causes the anthracnose disease in common bean plants (*Phaseolus vulgaris*). As a distinctive feature of several *Colletotrichum* species, *C. lindemuthianum* has a hemibiotrophic lifestyle, a nutrition/infection strategy consisting of a biotrophic phase in which it obtains nutrients from the host’s living cells, followed by a necrotrophic phase in which it kills and feeds on dead cells [1,2]. This necrotrophic phase is characterized by the secretion of plant-cell-wall-degrading enzymes (PCWDEs) [1,2,3,4]. The polysaccharides of the plant cell wall (PCW) include cellulose, hemicellulose, and pectin; thus, saprophytic and phytopathogenic fungi produce PCWDEs such as cellulases, hemicellulases, pectinases, and carboxy esterases, as well as accessory enzymes such as monooxygenases [1,2,3,4]. It has been reported that *C. lindemuthianum* isolates produce polygalacturonases, pectin lyases, pectate lyases, xylanases, β-xylosidases, β-glucosidases, α-glucosidases, mannanases, and cellulases in culture media with different substrates [5,6,7,8,9,10,11,12,13,14].

The interaction of *C. lindemuthianum*-*P. vulgaris* has led to a process of coevolution, where both species show intraspecific diversification through the generation of races or fungal pathotypes with variable virulence against bean varieties with different levels of resistance [15,16]. A great diversity of pathotypes has been identified among several fungal isolates collected from different bean-growing areas of the world using a system of 12 differential bean cultivars [15,16,17,18]. Worldwide, 298 pathotypes distributed in 29 countries have been identified, 64 of which are found in Mexico [15,16,19,20]. Interestingly, several isolates identified as pathotype 0 (P0) failed to infect all 12 differential bean cultivars, suggesting a preference for a saprophytic lifestyle [15,16]. This diversity of pathotypes ranging from high to low pathogenicity, and saprophytic behavior among *C. lindemuthianum* individuals is consistent with the idea that pathogenic fungi have high genomic plasticity for a rapid adaptation to host range, host evolution, and environmental changes [21,22].

Currently available genomic, transcriptomic, and proteomic information has revealed that fungi secrete a number of diverse carbohydrate active enzymes (CAZymes), which are associated with their nutritional strategies and host specificities [23]. For instance, it has been reported that many species of plant pathogens exhibit greater xylanase activity than does the saprophytic fungus *Trichoderma ressei* when it is evaluated on grass cell walls [24]. Furthermore, an evaluation of different fungal pathogenic species revealed greater hydrolysis than nonpathogenic fungal species when they interacted with filter paper, arabinoxylan, alfalfa, soybean steam, corn stalks, and switchgrasses [25]. In general, these studies have focused on interspecific differences using an individual representing each species of interest. However, the intraspecific capacity of phytopathogenic fungi to degrade PCW is unknown. In this sense, *C. lindemuthianum* is an excellent model for exploring this topic.

Previously, we found that two pathotypes of *C. lindemuthianum* (pathogenic P1472 and non-pathogenic P0) showed differential gene expression and/or secretion levels of pectin lyase, endo-1,4-β-D-xylanase, and β-galactosidase in cultures with pectin, polygalacturonic acid, arabinogalactan, xylan, and PCW from *P. vulgaris* as carbon sources, suggesting a relationship with their lifestyle differences [11,12,14,26]. Furthermore, these findings suggest the differential production of PCWDEs with biotechnological potential. In this study, we report on the differential capacity for mycelial growth, carbon catabolite repression, hemicellulolytic and cellulolytic activity, and PCWDE secretion in four *C. lindemuthianum* pathotypes with different virulence levels and lifestyles in cultures supplemented with glucose or natural plant substrates.

## 2. Materials and Methods

### 2.1. Strains and Culture Conditions

As mentioned above, the characterization of pathotypes was carried out using the system of 12 differential bean cultivars (Appendix A) that has been applied worldwide for *C. lindemuthianum* isolates [15,16,18]. Thus, the virulence level or virulence index (VI) of each pathotype is calculated using the number of susceptible bean cultivars ×100/total number of differential cultivars [16]. In this study, we used four pathotypes of *C. lindemuthianum* isolated from bean crops in different regions of Mexico (Appendix A). Pathotypes P0 (VI = 0.0), P1088(VI = 16.6), and P1472 (VI = 33.3) were collected from bean crops in central and northern Mexico [15]. Pathotype P2395 (VI = 58.3) was isolated from bean crops in central Mexico and kindly provided by Dra. Brenda Z. Guerrero-Aguilar, Dr. José L. Pons-Hernández, and Dr. Raul Rodríguez-Guerra from Centro de Investigación Regional del Centro-INIFAP, Celaya, Mexico. Fungi were maintained on a potato dextrose agar (PDA) medium prepared according to French and Hebert [27]. For the analysis of the time course of mycelial growth and enzymatic activity, four 5 mm diameter blocks of 15-day-old mycelium of each pathotype grown in PDA at room temperature were inoculated into 125 mL Erlenmeyer flasks containing a Mathur minimal medium (MMM), with glutamate as the preferred nitrogen source for the growth of this fungus, according to Acosta-Rodríguez et al. [10]. Treatments for each pathotype consisted of MMM supplemented either with 2.5% glucose (BD, Bioxon, Ciudad de México, Mexico), fresh and ground bean hypocotyls or green beans of *P. vulgaris,* or dried and ground water hyacinth (*Eichhornia crassipes*) as a carbon source. The cultures were shaken at 115 rpm for 16 days at 18 °C, under natural light and darkness conditions.

### 2.2. Enzyme Assay and Protein Quantitation

Assays of the enzymatic activity of α-L-arabinofuranosidase (ABF), β-(1,4)-D-xylanase (XYL), β-xylosidase (XYLO), and 1,4-β-cellobiohydrolase (CBH) were performed every 24 h. The extracellular medium (EM) of the cultures was separated by vacuum filtration using 3 mm Whatman filter paper and used to determine protein and enzymatic activity, while the mycelium was used to determine growth by dry weight after drying at 65 °C for 24 h.

The enzyme activities of ABF, XYLO, and CBH were measured by a fluorogenic method using 4-methylumbelliferyl-α-L-arabinofuranoside (4-MU-ABF) or 4MU-xilopiranoside (4MU-Xyp) or 4MU-β-D-cellobioside (4MU-C), respectively (Sigma-Aldrich St. Louis, MO, USA). The reaction mixtures contained 5 μM 4-MU-ABF or 25 μM 4MU-Xyp or 5 μM 4MU-C; 50 μL of EM; and 40 μL of 50 mM sodium acetate buffer, pH 5.0 (buffer A), in a final volume of 100 μL. The mixtures were incubated at 50 °C, and after 45–60 min, the reaction was stopped with 1 mL of a solution containing 0.5 M Na_2_CO_3_ and 0.1 N NaOH, pH 10.4. The released 4-MU was measured in 200 μL of the sample on a Varioskan Flash (Thermo Scientific, Waltham, MA, USA) with excitation and emission wavelengths set at 350 and 440 nm, respectively. The specific enzymatic activity was expressed as nanomoles of 4-MU released/min/μg protein and was converted to IU/mL. One international unit (IU) of enzyme activity corresponds to one micromole of 4-MU residues released/min/mg protein.

The enzyme activity of XYL was measured by a colorimetric method using xylan coupled to Remazol Brilliant Blue (RBB-xylan) (Sigma–Aldrich, St. Louis, MO, USA). The reaction mixtures contained 2.8 mg of RBB-xylan, 50 μL of EM, and 40 μL of buffer A in a final volume of 100 μL. The mixtures were incubated at 30 °C, and after 30 min, the reaction was stopped with 200 μL of 96% ethanol, and the absorbance was determined in 200 μL of the sample at 595 nm on a Varioskan Flash (Thermo Scientific, Waltham, MA, USA). Enzyme-specific activity was expressed as nanomoles of RBB released/min/μg protein and was converted to IU/mL. One international unit (IU) of enzyme activity corresponds to one micromole of RBB residues released/min/mg protein. For the sake of brevity, all data referring to specific activities in the text will be given without further definition.

The protein concentration in each sample was determined by Bradford’s method using a Coomassie Protein Assay Kit, according to the manufacturer’s instructions (Thermo Scientific, Waltham, MA, USA), and the levels of absorbance were detected on a Varioskan Flash (Thermo Scientific, Waltham, MA, USA).

### 2.3. Statistical Analysis

Significant changes in mycelial growth (dry weight) and extracellular enzyme activity among the pathotypes of *C. lindemuthinaum* were analyzed through Student’s *t* test, where *p* values < 0.05 were considered significant. One-way ANOVA for each substrate and two-way ANOVA (four pathotypes and four carbon sources) were performed for each day of incubation. In addition, different values of substrates and pathotypes were compared using the Tukey post hoc test. All the statistical analyses were performed using JMP Pro 13 Statistical Discovery LLC software (SAS, Institute Inc., Cary, NC, USA).

### 2.4. Secretome Analysis

Sample preparation was carried out as follows: four blocks of 5 mm of diameter of mycelia of each pathotype grown in PDA for 15 days were inoculated into 125 mL Erlenmeyer flasks containing 50 mL of a potato dextrose (PD) medium and shaken at 18 °C and 115 rpm. After seven days, the mycelia were separated by vacuum filtration using 3 mm Whatman filter paper, washed with sterile distilled water, and transferred to 125 mL Erlenmeyer flasks with 50 mL of MMM supplemented with 2.5% glucose or green beans as the carbon source. Cultures were shaken at 18 °C and 115 rpm for 10 days, as after this time, the highest amount of total protein secretion was detected by time course analysis (Appendix A).

### 2.5. Sample Preparation and Processing

The EM of each culture obtained by vacuum filtration was concentrated in a Pierce Protein Concentrator PES 10K MWCO, by centrifugation of 5–20 mL at 3260× *g* for 40 min at room temperature (Thermo Scientific, Waltham, MA, USA). Next, 0.5 mL of the supernatant was again concentrated in an Amicon Ultracel-10k Ultra centrifugal filter unit at 14,000× *g* and 4 °C for 30 min (Merck Millipore Ltd., Darmstadt, Germany), and samples with total protein concentrations ranging from 152 to 308 µg/mL in volumes ranging from 1 to 1.5 mL were obtained. The protein concentration in each sample was determined by Bradford’s method using a Coomassie Protein Assay Kit, according to the manufacturer’s instructions (Thermo Scientific, Waltham, MA, USA), and the levels of absorbance were detected on a Varioskan Flash (Thermo Scientific, Waltham, MA, USA). In total, eight samples were prepared, two per pathotype—one after glucose treatment and the other under green bean treatment.

For proteomic analysis, the samples were digested with trypsin, identified, and quantified using a Nano LC-MS/MS platform (Thermo Scientific, Waltham, MA, USA) through the service of Creative Proteomics, Inc. (New York, NY, USA) Protein extraction was performed by methanol and chloroform precipitation and centrifugation at 12,000 rpm at 4 °C for 15 min. Precipitates were then boiled for 10 min, separated by SDS–PAGE gel (12%), and silver-stained. Subsequently, the samples were subjected to gel digestion. The gel slice from each gel cut into 1 mm^3^ cubes was treated with 1 mL 30 mM K_3_Fe (CN)_6_/100 mM Na_2_S_2_O_3_ (1:1, *v*/*v*), until the brown disappeared, the supernatant was removed, and the reaction was stopped adding 200 μL of water, the supernatant was discarded after 10 min. Then, 1 mL of 50 mM ammonium bicarbonate was added, and after 30 min, the supernatant was removed, and 500 μL of acetonitrile (ACN) was added and incubated for 30 min. ACN was removed, and the gel slice was rehydrated in 10 mM DL-dithiothreitol (DTT) by incubating at 56 ℃ for 1 h. After the DTT was removed, 500 μL of ACN was added and incubated for 10 min at room temperature. The ACN was removed, and 50 mM iodoacetamide (IAA) was added to cover the gel slice and incubated for 30 min at room temperature in the dark. IAA was removed, and 500 μL of ACN was added to cover the gel slice and incubated for 10 min at room temperature. A trypsin digestion solution was added to the gel slices, incubated on ice for 45 min, and then incubated overnight at 37 ℃. The supernatant was recovered, and 50 mM of an ammonium bicarbonate/acetonitrile solution (1:2, *v*/*v*) was added to the gel slices and incubated for 1 h at 37 ℃. Next, 1% trifluoroacetic acid (TFA) was added and incubated for 30 min, and then 50 mM of the ammonium bicarbonate/acetonitrile solution (1:2, *v*/*v*) was added. Extracts of each sample were combined, and the extracted peptides were lyophilized to near dryness. Extracted peptides were resuspended in 20 μL of 0.1% formic acid before LC-MS/MS analysis.

### 2.6. LC-MS/MS Analysis

Nano LC-MS/MS analysis was performed on a UPLC Nanoflow: Ultimate 3000 nano UHPLC system (Thermo Fisher Scientific Waltham, MA, USA), Nanocolumn: trap column (PepMap C18, 100 Å, 100 μm × 2 cm, 5 μm), and an analytical column (PepMap C18, 100 Å, 75 μm × 50 cm. 2 μm). Loaded sample volume: 1 μg. Mobile phase: A: 0.1% formic acid in water; B: 0.1% formic acid in 80% acetonitrile. Total flow rate: 250 nL/min. LC linear gradient: from 2 to 8% buffer B in 3 min, from 8% to 20% buffer B in 56 min, from 20% to 40% buffer B in 37 min, then from 40% to 90% buffer B in 4 min.

### 2.7. Mass Spectrometry

Mass spectrometry was performed with a Q Exactive HF mass spectrometer (Thermo Fisher Scientific, Waltham, MA, USA) with an electrospray ionization (ESI) nanospray source. The full scan was performed at 300–1650 *m*/*z* at a resolution of 60,000 at 200 *m*/*z*; the automatic gain control target for the full scan was set to 3 × 10^6^. The MS/MS scan was operated in Top 20 mode using the following settings: resolution 15,000 at 200 *m*/*z*; automatic gain control target 1e5; maximum injection time 19 ms; normalized collision energy at 28%; isolation window of 1.4 Th; charge state exclusion: unassigned, 1, >6; and dynamic exclusion 30 s, performed using JMP Pro 13 Statistical Discovery LLC software (SAS, Institute Inc., Cary, NC, USA).

### 2.8. Data Analysis

For data analysis, eight raw MS files were analyzed and searched against the *Colletotrichum* protein database (UniProt) based on the species of the samples using MaxQuant (1.6.2.6) [28]. The parameters were set as follows: the protein modifications were carbamidomethylation (C) (fixed), oxidation (M) (variable); the enzyme specificity was set to trypsin; the maximum missed cleavages were set to 2; the precursors ion mass tolerance was set to 10 ppm; and MS7MS tolerance was 0.5 Da. Of the differentially expressed proteins, a quantitative ratio over 1.5 was considered up-regulation, while a quantitative ratio less than 1/1.5 was considered down-regulation.

A heatmap visualization of the secreted PCWDEs was constructed with GraphPad software v10.0.3 (GraphPad PRISM).

## 3. Results

### 3.1. Fungal Growth

Glucose is proposed to be the preferred carbon source because it is energetically most suitable for survival, rapid growth, and development during the colonization of several habitats [29,30]. This proposed preference implies that all pathotypes of *C. lindemuthianum* should grow better in cultures supplemented with glucose than in those supplemented with other carbon sources. To test this idea, we evaluated mycelial growth in cultures supplemented with glucose as the sole carbon source as well as media containing natural plant substrates of *P. vulgaris* with different percentages of pectin, hemicellulose, and cellulose such as bean hypocotyls (23.14% hemicellulose and 30.64% cellulose, dry weight) and green beans (47–50% pectin, 21–24% hemicellulose, and 16–21% cellulose, fresh weight, depending on the bean variety) [31,32]. Additionally, we used water hyacinth (*E. crassipes*), whose plant tissue is rich in hemicellulose (48% hemicellulose, 20% cellulose and 3.5% lignin) [33].

In cultures with glucose, the mycelial growth of the four pathotypes differed (Figure 1A). Pathotypes P0 and P1472 showed low growth, reaching maximums of 129 and 53 mg (dry weight) at 12 days, respectively. By contrast, P1088 and P2395 reached maximum concentrations of 311 and 255 mg, respectively, at nine days. In particular, P1088 and P2395 showed greater growth after the third and sixth days of incubation, respectively, than did P0 and P1472. The ANOVA results showed significant differences among P0, P1088, and P2395, and this corroborated the finding that P1088 exhibited the greatest growth and P1472 exhibited the least growth (Appendix A).

In cultures with bean hypocotyls, all pathotypes grew better than in those grown with glucose (Figure 1B). However, the growth profiles were different. Thus, P1088 and P2395 reached maximums of 254 and 260 mg, respectively, at seven days, while P0 and P1472 reached maximums of 280 and 257 mg at 16 days, respectively. ANOVA revealed significant differences among P1088 and the other pathotypes, and pathotypes P1088 and P0 exhibited the greatest and least growth, respectively (Appendix A).

In cultures with green beans, P0 and P1472 showed a rapid growth between three and four days, reaching maximums of 340 and 316 mg, respectively, which were greater than those of cultures grown with glucose and bean hypocotyls in this same period (Figure 1C). By contrast, P1088 and P2395 showed lower and slower growth as compared with cultures grown with glucose and bean hypocotyls, reaching optimal values of 157 and 173 mg after eight and 16 days, respectively. ANOVA showed significant differences among the four pathotypes, confirming that P0 and P1472 exhibited the best growth (Appendix A).

Finally, water hyacinth was the best carbon source for the growth of all pathotypes (Figure 1D). The mycelial masses of P0 and P1472 rapidly increased from the first day in the range of 400 to 500 mg, which was maintained for eight days. Subsequently, the two pathotypes reached maximums of 639 and 625 mg at nine days, respectively. However, P1472 reached a maximum of 895 mg at 16 days. Pathotypes P1088 and P2395 showed lower growth (with maximums of 313 and 317 mg, respectively); however, this growth was greater than that of cultures with glucose and with hypocotyls. ANOVA revealed significant differences among P1088, P1472, and P2395 and that P0 and P1472 had the best growth (Appendix A). In addition, the growth data showed significant differences among all the substrates, and water hyacinth was the best substrate, followed by bean hypocotyls (Appendix A).

### 3.2. Endo-β-1,4-xylanase Activity

Xylan is the second most abundant polysaccharide in PCW and consists of a backbone of β-(1,4)-D-xylose that can be branched by different glycosidic residues, such as L-arabinofuranose, L-arabinopiranose, L-rhamnose, D-glucuronic acid, and D-galacturonic acid, which can bind to the ends of the pectic polysaccharides and cellulose fibers [34,35]. In cultures with glucose as the carbon source, P0 and P1472 showed low levels of XYL activity (Figure 2A), in agreement with previous reports [14,26] and consistent with the CCR of hydrolytic enzyme-encoding genes in filamentous fungi [36,37,38].

However, the levels of XYL activity were greater in P1088 and P2395 (with maximums of 32 and 26 IU/mL, respectively) than in P0 and P1472 (with maximums of 16 and 9 IU/mL, respectively), suggesting a differential CCR (Figure 2A). ANOVA revealed significant differences among P0, P1472, and P2395, with the latter exhibiting the greatest XYL activity (Appendix A).

In cultures with bean hypocotyls (Figure 2B), pathotypes P0, P1088 and P2395 showed maximum enzymatic levels between the second and third days. However, the enzyme activity was variable. Thus, P0 showed the highest activity (74 IU/mL) on the second day, while P2395 and P1088 had slightly lower activity levels (66 and 64 IU/mL, respectively) on the third day of incubation. By contrast, P1472 showed a maximum activity (26 IU/mL) after seven days. There were no significant differences among pathotypes, and P1088 exhibited the best XYL activity (Appendix A).

In cultures with green beans (Figure 2C), the enzymatic profiles were different and, except for P2395, lower than those in cultures with bean hypocotyls. The best XYL activity was that of P2395, with an increase from the first day, and the activity peaked at four, nine, and fourteen days (with maximums of 37, 52, and 59 IU/mL, respectively). P0 and P1472 showed the greatest activity on the fifth and second days, with values of 42 and 35 IU/mL, respectively. P1088 produced XYL activity similar to that of glucose cultures. There were significant differences between P2395 and the other pathotypes, and this pathotype had the greatest XYL activity (Appendix A).

We detected the greatest amount of XYL activity in cultures supplemented with water hyacinth (Figure 2D). P0 showed the greatest activity (102 IU/mL) at six days, followed by P1472 (83 IU/mL) after four days, while P2395 and P1088 showed lower levels of activity, with maximums of 65 IU/mL on the seventh day and 35 IU/mL on the first day, respectively. ANOVA revealed significant differences among P1088, P1472, and P2395 and demonstrated that P1472 had the greatest XYL activity and was the best carbon substrate for enzyme induction (Appendix A).

### 3.3. α-L-Arabinofuranosidase Activity

Debranching enzymes allow access to enzymes that degrade the backbones of pectin and hemicellulose polysaccharides, releasing secondary substituents, such as monosaccharides or oligosaccharides. ABF activity releases lateral residues of α-L-arabinofuranose linked at the C3 or C2 of xylan and pectin [39,40,41]. In cultures with glucose, ABF activity was very low for all pathotypes, corroborating CCR (Figure 3A), and there were no significant differences among them (Appendix A).

In cultures with bean hypocotyls (Figure 3B), P1088 showed an increase in activity between four and five days, with a maximum of 62 IU/mL after five days. Pathotype P2395 exhibited sustained high activity between six and nine days, with a maximum of 47 IU/mL on the sixth day, while P0 and P1472 exhibited the lowest ABF activity. ANOVA revealed significant differences among P0, P1088, and P2395, with the latter exhibiting the greatest ABF activity (Appendix A). In cultures with green beans (Figure 3C), P2395 had the best ABF activity, with a maximum of 64 IU/mL after five days of incubation; however, the enzymatic profile of this pathotype showed several peaks during the incubation period. There were significant differences among all pathotypes, and P2395 exhibited the best ABF activity (Appendix A).

In cultures with water hyacinth (Figure 3D), all pathotypes showed similar enzymatic profiles, with the highest levels of production occurring between four and eight days. Thus, P2395 produced the highest ABF activity (72 IU/mL) after seven days, P1472 produced the highest ABF activity (43 IU/mL) at four days, and P1088 produced the highest ABF activity (40 IU/mL) after six days, while P0 showed a slight increase at four days. ANOVA revealed significant differences between P0 and P1088, with the latter exhibiting the greatest ABF activity (Appendix A). In addition, there were significant differences among glucose, green hypocotyls, and green beans, with the latter being the best substrate, followed by water hyacinth (Appendix A).

### 3.4. β-xylosidase Activity

XYL hydrolyzes the β-1,4 bond of xylan, generating xylooligosaccharides, which are further hydrolyzed by XYLO to xylose units [42]. In cultures with glucose, the levels of XYLO activity were very low or undetectable for pathotypes P0, P1088, and P1472, suggesting a strong CCR; however, P2395 showed a slight increase of 12.88 IU/mL at 14 days (Figure 4A).

Apart from this slight increase, there were no significant differences among the pathotypes (Appendix A). In cultures with bean hypocotyls (Figure 4B), only two pathotypes showed an increase in XYLO activity. Pathotype P0 produced 35 IU/mL on the second day, followed by a further increase between 11 and 16 days, with a maximum of 27 IU/mL after 14 days, while P2395 produced 20 IU/mL on the fifth day. ANOVA revealed significant differences between P0 and the other pathotypes, and P0 was the best XYLO producer (Appendix A). In cultures with green beans (Figure 4C), P0, P2395, and P1472 showed increases in activity. P2395 produced a maximum of 25 IU/mL on the third day, while P0 and P1472 showed slight increases between the seventh and eighth days and maximums of 21 and 13 IU/mL, respectively, after 14 days. ANOVA revealed significant differences among P0, P1088, and P1472, with the latter being the best XYLO producer (Appendix A).

In cultures with water hyacinth (Figure 4D), only P0 and P1472 produced high XYLO activity. Accordingly, P1472 produced 36 IU/mL on the sixth day and 57 IU/mL on the eighth day of incubation. P0 produced 34 IU/mL on the sixth day and 41 IU/mL after 12 days of incubation. ANOVA revealed significant differences among P0, P1088, and P1472, with the latter being the best XYLO producer (Appendix A). In addition, the ANOVA showed significant differences among glucose, green hypocotyls, and water hyacinth, with the latter being the best carbon substrate for XYLO production (Appendix A).

### 3.5. 1,4-β-cellobiohydrolase Activity

Little is known about the cellulolytic activity of *C. lindemuthianum* [10]. Cellulose 1,4-β-cellobiohydrolase (reducing end) (CBH1) attacks the available cellulose ends, which are further degraded by β-glucosidase to glucose [43,44]. As expected, due to CCR, CBH1 activity was very low or basal for all pathotypes in glucose cultures (Figure 5A).

However, P2395 produced a peak at 22 IU/mL on day 14; nevertheless, ANOVA did not reveal significant differences among the pathotypes (Appendix A). In cultures with bean hypocotyls (Figure 5B), P0 had an activity peak of 26 IU/mL on the second day, and P1472 had a peak of 21 IU/mL on the third day. Moreover, P1088 showed a peak activity of 31 IU/mL and P2395 showed a peak activity of 53 IU/mL after six and 16 days of incubation, respectively. The rest of the time, the activity levels were as low as those in the glucose cultures. ANOVA did not reveal significant differences among pathotypes (Appendix A).

When green beans were used as a carbon source (Figure 5C), P0 and P1088 had increased activity on the second and fifth days, with maximum activity values of 36 IU/mL and 29 IU/mL, respectively, while P1472 and P2395 had lower activity. ANOVA revealed significant differences among P0, P1088, and P1472, and P1088 had the greatest CBH1 activity (Appendix A).

Finally, in cultures with water hyacinth (Figure 5D), all pathotypes produced basal enzyme levels during the first three days and between 10 and 16 days. From four to nine days, the activity increased slightly in P0 (18 IU/mL) and P2395 (20 IU/mL); P1088 showed a peak activity (24 IU/mL) after nine days, and P1472 produced two peaks (37 IU/mL and 39 IU/mL) on the sixth and eighth days, respectively. ANOVA revealed significant differences among P1088, P1472, and P2395, and P1472, with the latter exhibiting the greatest CBH1 activity (Appendix A). ANOVA revealed significant differences among glucose, green beans, and water hyacinth, with green beans being the best carbon substrate for CBH1 production (Appendix A).

### 3.6. Secreted PCWDEs

A total of 59 PCWDEs active toward pectin, hemicellulose, and cellulose, belonging to 30 CAZy families, were identified with reference to databases of the genus *Colletotrichum* (Table 1). Most of these PCWDEs were previously identified only by homology in the genomes available for the genus (Appendix A). Among the 59 identified enzymes, 15 were pectinases, 19 were hemicellulases, 10 were debranching enzymes, 12 were cellulases, and 3 belonged to auxiliary activities (AAs) (Table 1 and Table 2). Debranching enzymes have activity on pectin and hemicellulose, and AAs have activity on cellulose.

Hemicellulases were the most abundant, highlighting the secretion of β-xylanases and mannosidases (Table 1 and Table 2, Figure 6A,C). No β-xylosidases were detected, but a bifunctional β-xylosidase/arabinofuranosidase, GH43, was identified (Table 1, Appendix A). However, the possibility of the secretion of other enzymes before 10 days should be considered.

Pectinases were also abundant, with high percentages of pectate lyases and endopolygalacturonases (Table 1 and Table 2, Figure 6A,B). The debranching enzymes mainly comprised arabinofuranosidases, α- and β-galactosidases, and feruloyl esterases (Table 1 and Table 2, Figure 6A,D). Cellulases were less abundant than pectinases and hemicellulases, highlighting the secretion of beta-glucosidases, cellobiohydrolases, and glucosyl hydrolases of family 131 (Table 1 and Table 2, Figure 6E), and the AAs detected were cellobiose dehydrogenases (Table 1 and Table 2, Figure 6E). Additionally, we identified hemicellulases and debranching enzymes, such as XYL GH11, and EBG GH30, and the pectinase PEL PL1, encoded by the *xyl1, ebg,* and *Clpnl2* genes, respectively, which were previously isolated from P0 and P1472 [12,14,26].

The results revealed differential secretion and regulation of PCWDEs among the pathotypes, and there were more upregulated enzymes in glucose cultures than in green bean cultures (Figure 7, Appendix A). In general, in all pathotypes in glucose cultures, the most upregulated enzymes were mainly pectinases and debranching enzymes. However, P0 and P1472 had more upregulated pectinases, hemicellulases, and debranching enzymes than did P1088 and P2395 (Figure 7). The upregulated pectinases were mainly pectate lyases, rhamnogalacturonan acetyl esterases, rhamnogalacturonate lyase, endopolygalacturonase, and pectin lyase, and only one pectate lyase was downregulated. The upregulated hemicellulases and debranching enzymes were mainly xylanases, endo-1,3(4)-β-glucanases, α- and β-galactosidases, and a feruloyl esterase. One β-xylosidase/arabinofuranosidase and one α-L-arabinofuranosidase were downregulated. The following cellulases were also upregulated in some or all pathotypes: alpha/beta-glucosidases, β-glucosidases, glycosyl hydrolases family 131, and cellobiohydrolase CBH1. Only the cellobiohydrolases CBH2, a cellulase GH5, and a cellobiose dehydrogenase were downregulated in all pathotypes.

In green bean cultures, all the pathotypes showed few upregulated pectinases, mainly pectate lyases and one rhamnogalacturonan acetylesterase that was constitutive or upregulated (Figure 7, Appendix A). By contrast, there were several upregulated hemicellulases and debranching enzymes, mainly xylanases and mannosidases, xyloglucanase, β-galactosidase, α-L-arabinofuranosidase, and exo-alpha-(1->5)-L-arabinofuranosidase. Moreover, a GH10/CBM xylanase had the highest secretion by P2395. However, some xylanases, mannosidases and debranching enzymes were also downregulated, especially in the pathogens P1088, P1472, and P2395. One feruloyl esterase showed a different profile for each pathotype—constitutive, upregulated, or downregulated. Pathotype P0 had the greatest numbers of hemicellulases and debranching enzymes, while P1088 had the lowest numbers. In addition, β-xylosidase/arabinofuranosidase was upregulated only in P0, while exo-alpha-(1->5)-L-arabinofuranosidase was upregulated in P0, P1472, and P2395 and downregulated in P1088.

Overall, more cellulases were upregulated in P0 and P1088 than in P1472 and P2395. The cellulases CBH1 and CBH2 and the cellulase GH5 were upregulated for all pathotypes, while a cellobiose dehydrogenase was upregulated only for P0 and P1088. In addition, the enzymes GH131, one alpha/beta-glucosidase, and one cellobiose dehydrogenase were downregulated in all the pathotypes.

## 4. Discussion

Most filamentous fungi can metabolize different carbon sources, and it is widely accepted that CCR represses PCWDE genes to utilize a more readily available carbon sources, such as glucose present in the culture medium [29,45]. According to studies in filamentous fungi, such as saprophytic *Saccharomyces cerevisiae*, *Aspergillus nidulans*, *Neurospora crassa*, and *Trichoderma reesei*, and the pathogen *Magnaporte oryzae*, regulators of carbon and nitrogen metabolism ensure the preferential use of glucose and ammonium, as well as gene repression for the use of less favored sources such as xylose and nitrate [46,47]. Furthermore, glucose is assumed to be the preferred carbon source because it is energetically most suitable for survival, rapid growth, and development during the colonization of several habitats [29,30]. However, *C. lindemuthianum* prefers glutamate as a nitrogen source [10], and the evaluation of four pathotypes with different virulence levels and lifestyles revealed that glucose as the carbon source in a culture medium is not necessarily suitable for optimal growth. The pathotypes showed differential mycelial growth, revealing that P1088 and P2395 grew better than P0 and P1472. We speculate that slower growth on glucose would confer on the fungus the in vivo ability to degrade complex polymers usually present in the PCW more efficiently than if glucose were present. It is unlikely that *C. lindemuthianum* and other fungi find free glucose in the plant. As a consequence, they may not have an efficient glucose transporter, or, alternatively, they also require inducing the expression of genes involved in glycolysis.

Evaluation of the enzymatic activity of PCWDEs secreted by the pathotypes revealed their association with the observed growth. Consistent with the CCR of PCWDEs in filamentous fungi [29,45], P0 and P1472 in glucose cultures had low basal activities of XYL, ABF, XYLO, and CBH1, corroborating previous reports on their basal enzyme production and gene expression [10,11,14,26]. However, the production of very low or nearly undetectable basal levels of XYLO and CBH1 activities suggests a stronger levels of CCR for these enzymes. In addition, P1088 and P2395 had much lower basal ABF, XYLO, and CBH1 activities; however, they showed high XYL activity, and P2395 showed a peak of XYLO activity, suggesting the lower levels of CCR of xylanolytic genes. It has been reported that different levels of CCR optimize the growth of *S. cerevisiae* populations in stable and variable environments [48]. In this sense, *C. lindemuthianum* pathotypes have different levels of CCR in response to glucose, which could optimize the growth in different tissues of their host. This implies that the species could have a consortium behavior in their interaction with bean crops.

The evaluation of pathotypes in cultures with plant carbon sources of their host *P. vulgaris* and of water hyacinth also revealed differential mycelial growth that, with some exceptions, was better than that with glucose. In general, all pathotypes were able to degrade any tissue and were mainly producers of XYL and ABF, with a low production of XYLO and CBH1. The group and activity of PCWDEs secreted by filamentous fungi are proposed to depend on their lifestyle [46]. In this sense, in agreement with previous findings in other pathogenic and saprophytic species [24,25], the pathogens P1088, P1472, and P2395, which were incubated in bean hypocotyls cultures, produced greater XYL activity and grew better than did P0. Nevertheless, regardless of their lifestyle, the pathotypes clearly showed different behaviors in glucose, bean hypocotyls, green beans, and water hyacinth cultures, revealing differential abilities to degrade PCW. Furthermore, P0 and P1472, showed the greatest XYL and XYLO activities and growth in water hyacinth cultures, suggesting the use of xylose as a carbon source. Water hyacinth is one of the most ecologically detrimental aquatic plants in the world due to its rapid propagation, ecological adaptability, and negative socioeconomic and health impacts [49,50,51]. Therefore, the enzymatic degradation of water hyacinth is an environmentally friendly alternative for its management and for biotechnological applications in the production of fungal enzymes and/or carbohydrates of industrial interest [52,53].

It has previously been reported that in addition to glucose, CCR can be induced by other monosaccharides, such as pentoses [45]. In *Aspergillus niger*, the hemicellulolytic regulator Xlnr and the arabinolytic regulator AraR control the release of D-xylose and L-arabinose and the pentose catabolic pathway (PCP) that metabolizes these sugars [54,55]. However, in this study, better XYL and XYLO activities were not always related to better growth among the pathotypes, suggesting differences in the uptake and metabolism of different monosaccharides and levels of CCR by pentoses. For instance, P1088 had the best XYL and XYLO activities and mycelial growth in water hyacinth cultures, but these activities were similar in glucose and green bean cultures, respectively. Moreover, P2395 showed the greatest growth in water hyacinth cultures but showed the greatest XYL and XYLO activities in green bean cultures. However, a higher production of ABF activity was not related to better mycelial growth, suggesting that L-arabinose is not a preferred carbon source. Pathotypes P1088 and P2395 had the greatest ABF activity in the bean hypocotyls and green beans cultures, respectively, while P0 and P1472 had the greatest ABF activity in the bean hypocotyls. Despite the low CBH1 activity detected among the pathotypes, P1088 and P1472 showed significant activity in green bean and water hyacinth cultures, respectively, while P0 and P2395 had the greatest activity in green bean and bean hypocotyl cultures, respectively.

Analysis of the *C. lindemuthianum* secretome allowed for the first detection of several PCWDEs previously identified only by homology in the genomes of *Colletotrichum* species. Additionally, we detected PCWDEs encoded by genes previously isolated from the pathotypes P0 and P1472, which coincided with their genetic expression and corroborated their secretion [12,14,26]. Although we did not measure pectinase activity in this study, we also reported the results of the identification and quantification of these enzymes. Previously, 58 genes encoding pectinases were reported in the genome of *C. lindemuthianum* strain 89 [56]. In this study, we detected the secretion of 15 pectinases, mainly pectate lyases and polygalacturonases, suggesting intraspecific differential CCR by glucose and differential regulation of gene expression against natural plant substrates. Our results agree with other studies on filamentous fungi, as the lowest number of secreted PCWDEs detected here compared to the large number of genes identified by other authors has been previously described in individuals of *Aspergillus terreus*, *T. reesei*, *Myceliophthora thermophila*, *N. crassa*, and *Phanerochaete chrysosporium* in cultures with different natural plant substrates [57].

In agreement with the high hemicellulolytic capacity of the pathotypes, hemicellulases were the most abundant enzymes—mainly XYLs and MANs, as well as the xylan and mannan debranching enzymes; these enzymes release metabolizable carbon sources, such as xylose, arabinose, mannose, and galactose. The XYLs belonging to the GH10 family were more abundant than the XYLs of the GH11 family, and as no XYLOs were detected, the xylose release can be explained by the activity of the GH10 XYLs. The GH11 family XYLs depolymerize the main backbone of xylan, releasing xylooligosaccharides, while the GH10 family XYLs are more active at releasing xylotriose, xylobiose, or xylose from xyloolygosaccharides [42,58,59]. In addition, xylose would be released by the bifunctional β-xylosidase/arabinofuranosidase identified in pathotype P0. Similar enzymes have been characterized previously in *T. reesei* and *Phanerochaete chrysosporium* [60,61]. One function of mannans is as a storage reserve in bean seeds [62,63]; therefore, their degradation by MANs of the GH5 and GH26 families and its debranching enzymes detected release mannose and galactose. Galactomannan is a common hemicellulose in the Leguminosae family, with a seed content of 1–38% (dry weight) [59]. The cellulases secreted were mainly beta-glucosidases, cellobiohydrolases, and glucosyl hydrolases of the GH131 family. According to the biochemical characterization of the GH131 enzymes from *Podospora anserina* and *Coprinopsis cinerea*, they have a bifunctional exo-β-1,3-/-1,6- and endo-β-1,4 activity toward β-glucan polysaccharides and cellulosic derivatives [64,65]. In addition, the enzymes detected with AAs were cellobiose dehydrogenases, whose substrates are cellobiose or cello-oligosaccharides [66].

The secretomes of glucose cultures revealed a greater number of PCWDEs than those of green bean cultures and a secretion profile exclusive of each pathotype with differential regulation, corroborating the different levels of CCR detected in the species. In addition, the greater number of secreted PCWDEs in glucose cultures does not seem to be related to the level of virulence or lifestyle because P0 and P1472 had more upregulated enzymes than did P1088 and P2395. The secretion of several hemicellulases and cellulases was consistent with the basal enzymatic activity of XYL, ABF, XYLO, and CBH1 shown by the pathotypes in glucose cultures and those described for other filamentous fungi [29,45]. All pathotypes grown in green bean cultures had few upregulated pectinases; however, it should be considered that the secretion of a greater number of pectinases may occur in the first days of incubation, when the degradation of the pectin found in the middle lamella of plant tissues and that associated with hemicellulose and cellulose on PCW is required [12,67]. We corroborated the differential hemicellulolytic capacity among the pathotypes in green bean cultures; however, the non-pathogen P0 showed more upregulated hemicellulases, debranching enzymes, and cellulases than did the pathogens P1088, P1472, and P2395, while P1088 showed lower amounts of hemicellulases and debranching enzymes. Therefore, our results suggest that the greater hydrolysis of pathogenic than non-pathogenic fungal species in interactions with different substrates observed in previous studies [25] could be influenced by the differential regulation of PCWDEs of the individuals used in the analyses. However, in P2395, the greater secretion of a GH10/CBM β-xylanase, in addition to other XYLs, may explain its greater XYL activity in green bean cultures. The CBH1 activity detected in all pathotypes was in accordance with the secretion of CBH1; the upregulated CBH2 and GH5 cellulases also contributed to cellulose degradation.

## 5. Conclusions

*C. lindemuthianum* pathotypes show different mycelial growth and enzymatic profile production of PCWDEs in glucose, bean hypocotyls, green beans, and water hyacinth cultures. Glucose is not the preferred carbon source for the best mycelial growth of all pathotypes. Furthermore, the differences in the basal enzymatic activities of hemicellulases and cellulases in glucose cultures, the greater amount of PCWDE secretion with glucose than with green beans, and a secretion profile exclusive to each pathotype suggest different levels of CCR by glucose in the species. The pathotypes show a high differential hemicellulolytic capacity in both glucose and green beans, with a high secretion of XYLs and MANs, as well as xylan and mannan debranching enzymes that release metabolizable carbon sources. Pathotypes have a differential capacity to degrade host and water hyacinth tissues, but better enzymatic activity and PCWDE secretion are not always related to better mycelial growth; they must have CCR by pentoses and differences in the uptake and metabolism of different monosaccharides and/or disaccharides. A higher production of ABF activity was not related to better mycelial growth, suggesting that L-arabinose does not play a primary role as a carbon source. We propose that different levels of CCR could optimize growth in different host tissues and allow for consortium behavior in interactions with bean crops.

## Figures and Tables

**Figure 1 jof-10-00406-f001:**
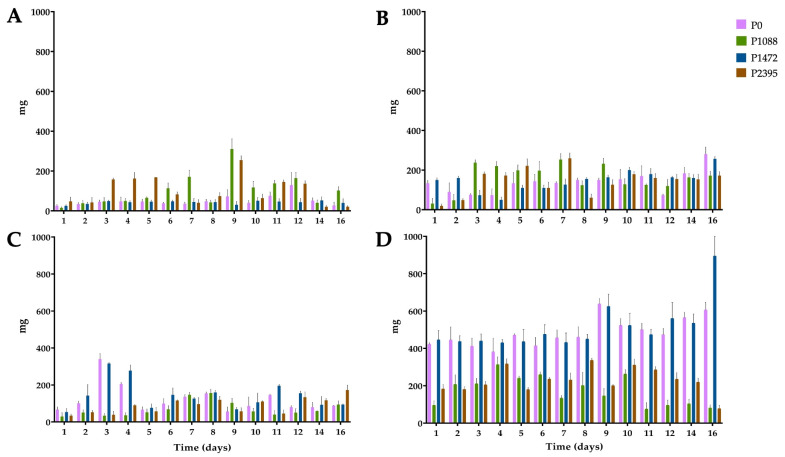
Growth of pathotypes P0, P1088, P1472, and P2395 of *C. lindemuthianum* in a modified Mathur medium, supplemented with different carbon sources: (**A**) glucose, (**B**) bean hypocotyls, (**C**) green beans, and (**D**) water hyacinth. Each bar shows the mean of triplicates ± SE.

**Figure 2 jof-10-00406-f002:**
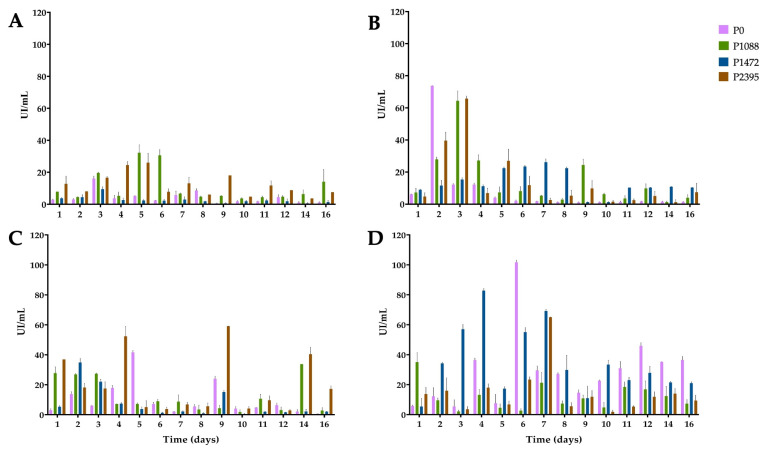
Endo-β-1,4-xylanase activity of pathotypes P0, P1088, P1472, and P2395 in a modified Mathur medium, supplemented with different carbon sources: (**A**) glucose, (**B**) bean hypocotyls, (**C**) green beans, and (**D**) water hyacinth. Each bar shows the mean of triplicates ± SE.

**Figure 3 jof-10-00406-f003:**
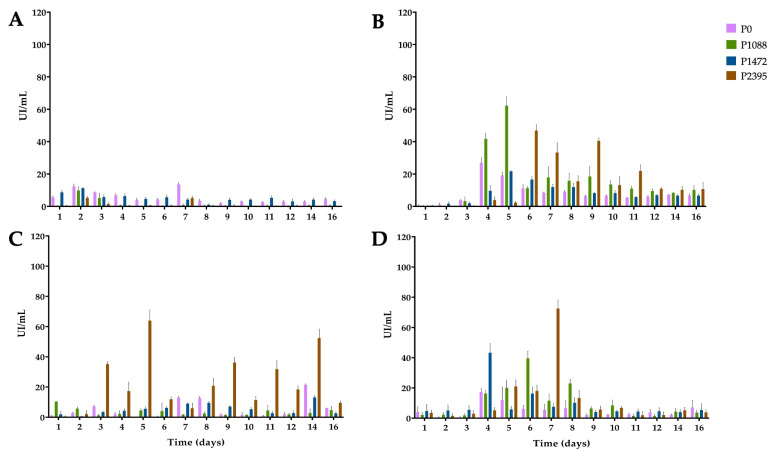
α-L-Arabinofuranosidase activity of pathotypes P0, P1088, P1472, and P2395 in a modified Mathur medium, supplemented with different carbon sources: (**A**) glucose, (**B**) bean hypocotyls, (**C**) green beans, and (**D**) water hyacinth. Each bar shows the mean of triplicates ± SE.

**Figure 4 jof-10-00406-f004:**
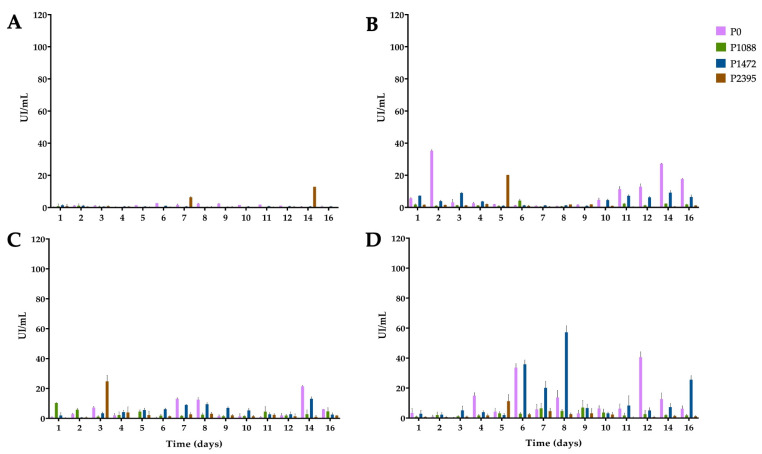
β-xylosidase activity of pathotypes P0, P1088, P1472, and P2395 in a modified Mathur medium, supplemented with different carbon sources: (**A**) glucose, (**B**) bean hypocotyls, (**C**) green beans, and (**D**) water hyacinth. Each bar shows the mean of triplicates ± SE.

**Figure 5 jof-10-00406-f005:**
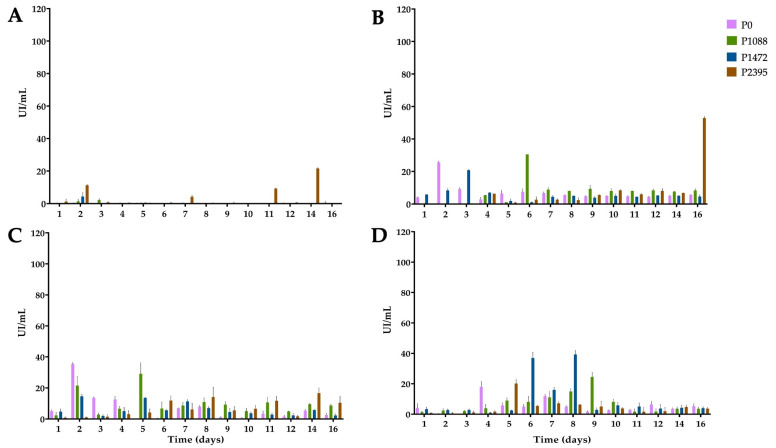
1,4-β-Cellobiohydrolase activity of pathotypes P0, P1088, P1472, and P2395 in a modified Mathur medium, supplemented with different carbon sources. (**A**) glucose, (**B**) bean hypocotyls, (**C**) Green beans, and (**D**) water hyacinth. Each bar shows the mean of triplicates ± SE.

**Figure 6 jof-10-00406-f006:**
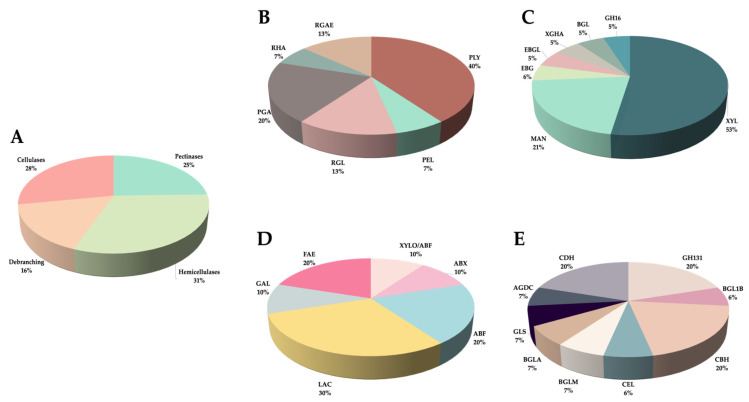
Percentages of PCWDEs identified in *C. lindemuthianum*. (**A**) Pectinases, hemicellulases, debranching enzymes, and cellulases (including AAs), (**B**) Pectinases, (**C**) Hemicellulases, (**D**) Debranching enzymes, and (**E**) Cellulases (including AAs). The abbreviations are listed in Table 1.

**Figure 7 jof-10-00406-f007:**
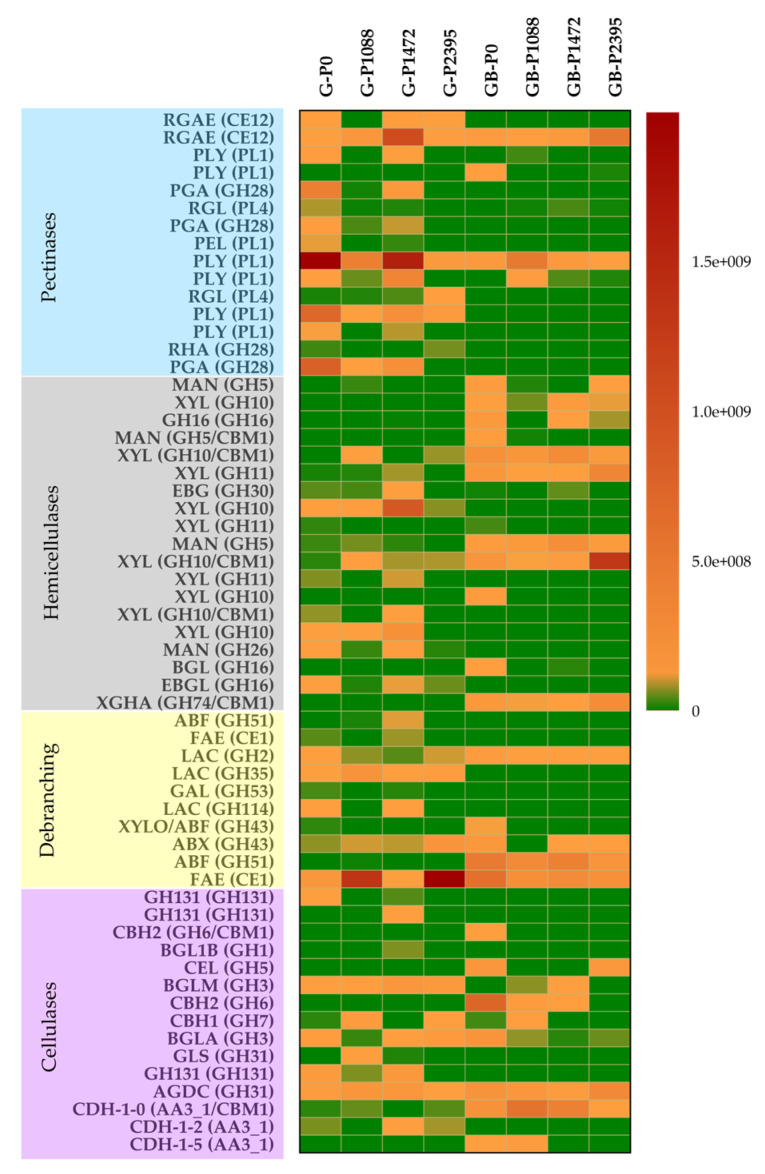
Heatmap representations of PCWDEs secreted by *C. lindemuthianum* pathotypes P0, P1088, P1472, and P2395 in glucose and green bean cultures. The pectinases, hemicellulases, debranching enzymes, and cellulases of each pathotype grown with glucose (G-P0, G-P1088, G-P1472, and G-P2395) or green beans of *P. vulgaris* (GB-P0, GB-P1088, GB-P1472, and GB-P2395) are shown. The scale of the heatmap ranges from high abundance (red color) to low abundance (green color). The abbreviations are listed in Table 1.

**Table 1 jof-10-00406-t001:** Secreted PCWDEs and their CAZy families identified in *C. lindemuthianum* pathotypes.

PCWDEs	Enzyme	# Genes	CAZy Family
**Pectinases**			
PLY	Pectate lyase	6	PL1
PEL	Pectin lyase	1	PL1
RGL	Rhamnogalacturonate lyase	2	PL4
PGA	Endopolygalacturonase	3	GH28
RHA	Alpha-L-rhamnosidase	1	GH28
RGAE	Rhamnogalacturonan acetylesterase	2	CE12
**Hemicellulases**			
XYL	Endo-1,4-beta-xylanase	4	GH10
XYL	Endo-1,4-beta-xylanase	3	GH10/CBM1
XYL	Endo-1,4-beta-xylanase	3	GH11
MAN	Mannan endo-1,4-beta-mannosidase	2	GH5
MAN	Mannan endo-1,4-beta-mannosidase	1	GH5/CBM1
MAN	Mannan endo-1,4-beta-mannosidase	1	GH26
EBG	Endo-beta-1,6-galactanase	1	GH30
EBGL	Endo-1,3(4)-beta-glucanase	1	GH16
XGHA	Xyloglucanase	1	GH74/CBM1
BGL	Beta-glucanase	1	GH16
GH16	Glycosyl hydrolase family 16	1	GH16
**Debranching enzymes**			
XYLO/ABF	β-xylosidase/arabinofuranosidase	1	GH43
ABX	Exo-alpha-(1->5)-L-arabinofuranosidase	1	GH43
ABF	Alpha-L-arabinofuranosidase	2	GH51
LAC	Beta-galactosidase	1	GH35
LAC	Beta-galactosidase	1	GH2
LAC	Alpha-galactosidase	1	GH114
GAL	Arabinogalactan endo-beta-1,4-galactanase	1	GH53
FAE	Feruloyl esterase	2	CE1
**Cellulases**			
BGLA	Beta-glucosidase	1	GH3
BGL1B	Beta-glucosidase	1	GH1
BGLM	Beta-glucosidase	1	GH3
CEL	Glycosyl hydrolase 5 family	1	GH5
CBH2	Cellobiohydrolase	1	GH6/CBM1
CBH2	Cellobiohydrolase	1	GH6
CBH1	Cellobiohydrolase	1	GH7
GLS	Alpha/beta-glucosidase agdC	1	GH31
AGDC	Alpha/beta-glucosidase agdC	1	GH31
GH131	Glycoside hydrolase family 131	3	GH131
**Auxiliary Activities**			
CDH-1-0	Cellobiose dehydrogenase	1	AA3_1/CBM1
CDH-1-2	Cellobiose dehydrogenase	1	AA3_1
CDH-1-5	Cellobiose dehydrogenase	1	AA3_1

**Table 2 jof-10-00406-t002:** Numbers of PCWDEs and CAZy families identified in each *C. lindemuthianum* pathotype.

Pathotype	# Genes	Pectinases	Hemicellulases	Debranching	Cellulases	Auxiliar Activities	CAZy Families
P0	50	11	19	7	12	3	26
P1088	33	8	12	5	8	2	24
P1472	43	11	13	8	11	2	25
P2395	27	7	10	5	5	2	20
Total	59	15	19	10	12	3	30

## Data Availability

Data are available within the article and Appendix A.

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
