# Peer review of "Differential Carbon Catabolite Repression and Hemicellulolytic Ability among Pathotypes of Colletotrichum lindemuthianum against Natural Plant Substrates"

_jof, 2024, doi:10.3390/jof10060406_

Round 1

Reviewer 1 Report

The manuscript by authors Díaz-Tapia et al. investigated the growth of four pathotypes of Colletotrichum lindemuthianum in four carbon sources (glucose, bean hypocotyls, green beans, and water hyacinth), the activities of four secreted cell wall degrading enzymes (endo-β-1,4-xylanase, α-L-Arabinofuranosidase, β-xylosidase, and 1,4-β-cellobiohydrolase), and the secreted cell wall degrading enzyme species. The significant differences were found among four pathotypes, suggesting that different levels of carbon catabolite regulation in the strains influence the differences in the growth of four pathotypes. This study can provide some basic references for the study about the carbon metabolism of C. lindemuthianum. The manuscript concluded that glucose is not a preferential carbon source for all pathotypes of C. lindemuthianum, differing from the existing general understanding. However, no plausible explanation of the molecular mechanism is provided for this conclusion. There have been a number of studies on the regulatory mechanisms of fungal carbon catabolite repression (CCR), including CCR-regulated genes affecting the expression of cell wall-degrading enzymes. Filamentous fungal CCR and carbon catabolite derepression (CCDR) are regulated by AMPK kinase Snf1, Pp4c phosphatase Smek1, and transcription factors CreA/Cre1 and Crf1. It is recommended that the authors compare the expression levels of these regulated genes in the four pathotypes of C. lindemuthianum using methods such as quantitative RT-PCR or RNA-seq. This will help to find out the causes that lead to the differences in the ability to utilize cell wall components and glucose among the four strains.

Line 253, in Figure 1, why did the biomass of the individual strains not increase with increasing incubation time?

Lines 28, 256, and 500, glucose is a preferred carbon source for fungi. Why did all strains grow slower in glucose medium?

Line 493, orizae should be revised to oryzae.

Author Response

Reviewer 1

Regarding review of our submission, below is the list of responses to your comments/queries. As you will notice, all suggestions were attended and duly replied. We hope that this contribution is now in an acceptable form for publication in the Journal of Fungi. We thank your consideration and peer review which will undoubtedly end in better communication. We will gladly consider further suggestions.

Major comments

The manuscript by authors Díaz-Tapia et al. investigated the growth of four pathotypes of Colletotrichum lindemuthianumin four carbon sources (glucose, bean hypocotyls, green beans, and water hyacinth), the activities of four secreted cell wall degrading enzymes (endo-β-1,4-xylanase, α-L-Arabinofuranosidase, β-xylosidase, and 1,4-β-cellobiohydrolase), and the secreted cell wall degrading enzyme species. The significant differences were found among four pathotypes, suggesting that different levels of carbon catabolite regulation in the strains influence the differences in the growth of four pathotypes. This study can provide some basic references for the study about the carbon metabolism of C. lindemuthianum. The manuscript concluded that glucose is not a preferential carbon source for all pathotypes of C. lindemuthianum, differing from the existing general understanding. However, no plausible explanation of the molecular mechanism is provided for this conclusion. There have been a number of studies on the regulatory mechanisms of fungal carbon catabolite repression (CCR), including CCR-regulated genes affecting the expression of cell wall-degrading enzymes. Filamentous fungal CCR and carbon catabolite derepression (CCDR) are regulated by AMPK kinase Snf1, Pp4c phosphatase Smek1, and transcription factors CreA/Cre1 and Crf1. It is recommended that the authors compare the expression levels of these regulated genes in the four pathotypes of C. lindemuthianum using methods such as quantitative RT-PCR or RNA-seq. This will help to find out the causes that lead to the differences in the ability to utilize cell wall components and glucose among the four strains.

Answer:

We thank this reviewer for carefully reading the manuscript and his very pertinent observations regarding the regulation of fungal carbon metabolism. AMPK kinase Snif1 and Pp4c phosphatase Smek 1 as well as the transcription factors he mentions play a central role in kinase cascades that act as metabolic sensors and regulate the expression of plant cell wall-degrading enzymes. This knowledge is well documented. To give the project a follow up, we considered important to know first the ability of C. lindemuthianum pathotypes to utilize different carbon sources and examine fungal growth and secreted CAZymes as a proteomic approach and a starting platform to our project. Understanding how the metabolism of carbon substrates is regulated at genomic level in our model is currently in progress. However, our results are very preliminar to include them in this submission. Hopefully, this will be the subject of a future paper.

Detail comments

Comment:

Line 253, in Figure 1, why did the biomass of the individual strains not increase with increasing incubation time?

Answer:

We have repeatedly observed this phenomenon. We believe that it may be due to the complexity of substrates added to cultures, which may need more than one enzyme to generate simple, assimilable products for cell growth. This and the transport of nutrients into the cell may slow down the growth and time-depending increase in cell biomass.

Comment:

Lines 28, 256, and 500, glucose is a preferred carbon source for fungi. Why did all strains grow slower in glucose medium?

 Answer:

There are other organisms that grow better in substrates other than glucose. For instance, we are currently working with Penicillium sp, a member of a fungal consortium. It grows better in sucrose and other oligomers than in glucose. We speculate that this would confer the fungus the in vivo ability to degrade complex polymers usually present in the plant cell wall more efficiently than if glucose were present. It is unlikely that C. lindemuthianum and other fungi find free glucose in the plant. As a consequence, they may not have an efficient glucose transporter or, alternatively, they also require inducing the expression of genes involved in glycolysis. We include this idea in the discussion (Lines 502-507).

Comment:

Line 493, orizae should be revised to oryzae.

Answer:

Ok. It was corrected.

Reviewer 2 Report

the discussion should be expanded

100 line: the provider of PDA should be mentioned

103: MM medium provider

90 line: table of cultivars

95 line: a table of pathotypes with detailed information should be provided

102 : what temperature? dark? light?

104: it should be explained which treatments are supplemented with which carbon sources. 

102 line 15 days, 107 line 16 days? Or was it a 15-day-old culture used for inoculation? 

157: what size of vacuum filtration, Watman? 

228: some citation in results? why ? 

still question about the origin of the isolate, 

416 It is not clear why their enzymes were selected and how they were detected. should be a clear statement in methods. now a lit lack.

supplementary: tables should be provided with all the information. detailed information.

Table S1. Table S2. S3  S4 S5 What do letters indicate in the table

S6-S7. As I assume, mark X means yes, the enzymes were found???

Author Response

Reviewer 2

Regarding review of our submission, below is the list of responses to your comments/queries. As you will notice, all suggestions were attended and duly replied. We hope that this contribution is now in an acceptable form for publication in the Journal of Fungi. We thank your consideration and peer review which will undoubtedly end in better communication. We will gladly consider further suggestions.

Major comments

Comment:

the discussion should be expanded

Answer:

Ok. The discussion has been expanded.

Detail comments

Comment:

Are all of the cited references relevant to the research?

No

yes, but some old publications used, could be used newer (not older than 10 year)

Answer:

All cited references are relevant to our research. In this area of knowledge, citations older than 10 years are direct antecedents of this work and/or contain basic information that continues to be essential.

Comment:

100 line: the provider of PDA should be mentioned

Answer:

Thank for this observation. PDA medium was not purchased. It was prepared according to French and Hebert (1982). This citation was included in the text (Lines 100-101) and the reference list.

Comment:

103: MM medium provider

Answer:

The MMM medium was not purchased. It was prepared according to Acosta-Rodríguez et al. (2005), reference [10]. This is now indicated in the text (Line 105).

Comment:

90 line: table of cultivars

Answer:

We did not include this table because it is published in references [16] and [18] (Line 91). However, in response to the request, we include the table in Supplementary materials.

Comment:

95 line: a table of pathotypes with detailed information should be provided

Answer:

Ok. We agree. This information is in a table included in Supplementary materials.

Comment:

102 : what temperature? dark? light?

Answer:

Ok. The incubation temperature in MMM is indicated in lines 108-109.

Ok. Information of the light and dark incubation conditions was included in line 109.

Comment:

104: it should be explained which treatments are supplemented with which carbon sources.

Answer:

Ok. An improved description is now included in lines 105-106.

Comment:

102 line 15 days, 107 line 16 days? Or was it a 15-day-old culture used for inoculation?

Answer:

Ok. 15-day-old cultures were used for inoculation. Phrasing was corrected (Line 102).

Comment:

157: what size of vacuum filtration, Watman?

Answer:

Ok. This information is included in the text (Lines 160-161).

Comment:

228: some citation in results? why ?

Answer:

Ok. The theoretical approach we describe is supported by these citations.

Comment:

still question about the origin of the isolate,

Answer:

The origin of the pathotypes is described in lines 93-99. Additionally, we included a table with detailed information in Supplementary Materials.

 Comment:

416 It is not clear why their enzymes were selected and how they were detected. should be a clear statement in methods. now a lit lack.

Answer:

Ok. The enzymes were obtained and identified by proteomic analysis (secretome). The methodological description of the preparation and analysis of secretomes is found in lines 156-229.

Comment:

supplementary: tables should be provided with all the information. detailed information.

Table S1. Table S2. S3 S4 S5 What do letters indicate in the table

Answer:

Thank for this observation. Values with different letters mean significant differences (p < 0.0001). We included this description in the tables.

Comment:

S6-S7. As I assume, mark X means yes, the enzymes were found???

Answer:

Ok. In table S8, the X indicate the enzymes detected in the secretome of each pathotype.

Ok. In table S9, the X indicate the enzymes upregulated in the secretome of each pathotype.

We included a detailed information.

Round 2

Reviewer 1 Report

This revised manuscript already meets the publication quality of the journal.

No comment.